# The Role of Platelets in Infective Endocarditis

**DOI:** 10.3390/ijms24087540

**Published:** 2023-04-19

**Authors:** Mustapha Abdeljalil Braï, Nadji Hannachi, Nabila El Gueddari, Jean-Pierre Baudoin, Abderrhamane Dahmani, Hubert Lepidi, Gilbert Habib, Laurence Camoin-Jau

**Affiliations:** 1IRD, APHM, MEPHI, IHU Méditerranée Infection, Aix Marseille University, 19-21 Boulevard Jean Moulin, 13005 Marseille, France; 2IHU Méditerranée Infection, Boulevard Jean Moulin, 13385 Marseille, France; 3Laboratoire de Biopharmacie et Pharmacotechnie, Faculté de Médecine, Université Ferhat Abbas Sétif I, Sétif 19000, Algeria; 4Service de Chirurgie Cardiaque, Hôpital de la Timone, APHM, Boulevard Jean-Moulin, 13385 Marseille, France; 5Service d’Anatomo-Pathologie, Hôpital de la Timone, APHM, Boulevard Jean-Moulin, 13385 Marseille, France; 6Service de Cardiologie, Hôpital de la Timone, APHM, Boulevard Jean-Moulin, 13385 Marseille, France; 7Laboratoire d’Hématologie, Hôpital de la Timone, APHM, Boulevard Jean-Moulin, 13385 Marseille, France

**Keywords:** infective endocarditis, platelets, *Staphylococcus aureus*, *Viridans streptococci*, *Enterococcus faecalis*

## Abstract

Over the last decade, the incidence of infective endocarditis (IE) has increased, with a change in the frequency of causative bacteria. Early evidence has substantially demonstrated the crucial role of bacterial interaction with human platelets, with no clear mechanistic characterization in the pathogenesis of IE. The pathogenesis of endocarditis is so complex and atypical that it is still unclear how and why certain bacterial species will induce the formation of vegetation. In this review, we will analyze the key role of platelets in the physiopathology of endocarditis and in the formation of vegetation, depending on the bacterial species. We provide a comprehensive outline of the involvement of platelets in the host immune response, investigate the latest developments in platelet therapy, and discuss prospective research avenues for solving the mechanistic enigma of bacteria–platelet interaction for preventive and curative medicine.

## 1. Introduction

Infectious endocarditis (IE) is a rare disease characterized by a bacterial or fungal infection of the endocardium. It is marked by valve destruction and the formation of vegetation, corresponding to a bacterial-infested clot of fibrin and platelets [1]. IE is a deadly disease, associated with difficult diagnosis, high morbidity, and both in-hospital and long-term mortality, which more frequently affects males than females. IE may occur both in patients with previous valve disease and on apparently normal valves and is particularly frequent in intravenous drug abusers [2]. The epidemiology of IE has progressively evolved over the past two decades. Healthcare-associated IE accounts for approximately 25% to 30% of contemporary cohorts, driven by an increase in the use of intravenous lines and intracardiac devices. Paradoxically, among the many species of pathogens identified in bacteremia, only a limited number of Gram-positive bacteria are responsible for IE.

Together, *Staphylococcus aureus*, *Streptococcus* spp., and enterococci are responsible for more than 80% of disease cases. These organisms have surface adhesins that allow them to attach to damaged valves [2]. The incidence of enterococcal endocarditis is likely to increase as the general population ages and the number of individuals at risk increases due to the prevalence of degenerative valve disease and genitourinary conditions in the elderly, which are often sources of enterococcal infection [3,4,5]. Although Gram-negative bacteria such as *Escherichia coli* are a common cause of bacteremia, they rarely cause endocarditis [6].

The occurrence of bacteremia on a pathological valve or foreign material results in complex interactions between the microorganism, the valve endothelium, and host immune responses. Normal valve endothelium is resistant to thrombosis and infection. However, its alteration due to native valve disease exposes the extracellular matrix components of the subendothelium which is thrombogenic and triggers platelet adhesion and activation, followed by rapid colonization of circulating microorganisms [1]. Differences in the affinity of microorganisms for the valve depend on adhesion molecules (MSCRAMM: microbial surface component reacting with adhesive matrix molecules). Fibrinogen- or fibronectin-binding proteins are present on the surface of microorganisms. Some microorganisms such as *S. aureus* can also bind directly to endothelial cells. The properties of these numerous adhesion molecules explain the predominance of Gram-positive cocci as the responsible microorganisms [7]. The particular adhesion capabilities of *S. aureus* may explain its predominance in endocarditis, occurring in the absence of pre-existing valvulopathy [2]. Adhesion of microorganisms is also promoted by integrin expression by endothelial cells in response to inflammation. After the adhesion phase, microorganisms can be internalized by endothelial cells where they can proliferate and diffuse or persist by evading antibiotics and the immune system. The mechanisms of colonization of the valve endothelium differ and are poorly understood for microorganisms with exclusively intracellular development, such as *Coxiella burnetii* and Bartonella.

Some microorganisms, particularly *S. aureus*, accompany their proliferation with biofilm formation leading to platelet aggregation embedded in a polysaccharide and protein network. Biofilm formation promotes virulence of the bacteria, particularly by protecting them from immune defenses and antimicrobial treatments [8]. Colonization of the fibrino-platelet thrombus by circulating microorganisms triggers an immune response, activating monocytes that secrete tissue factors and cytokines. The invasion of the valve tissue by microorganisms and then inflammatory cells is the cause of the main lesions and complications of IE. The vegetations correspond to a septic thrombus, whose growth is promoted by the activation of hemostasis by cytokines. Microorganisms that have colonized the vegetations are difficult to access by antibiotics and immune defenses [9].

Platelets play a key role during IE by combining their well-known hemostatic role with their proinflammatory and antibacterial powers [10]. Platelets are anucleate cytoplasmic fractions formed and released into the bloodstream by precursor cells known as megakaryocytes, which reside in the bone marrow. On their membrane surface, platelets express a variety of receptors that allow them to interact with host cells, including endothelial cells and leukocytes, plasma proteins, and pathogens. The platelets contain three main types of granules. The alpha granules, the largest and most significant in number, contain hundreds of bioactive proteins, including fibrinogen, von Willebrand factor (VWF). and chemokines, and their degranulation increases the surface area of the platelet membrane. The dense granules contain mainly bioactive amines, adenine nucleotides, and polyphosphates, as well as high concentrations of calcium. The lysosomes contain lysosomal enzymes such as beta hexosaminidase [11,12].

In this review, we discuss the role of platelets in IE, including the pathogenesis of vegetation, and the prognostic role of platelet parameters. The development of new approaches for diagnosis and research of IE might highlight the potential of targeting therapeutic candidates for the prevention and treatment of IE.

## 2. Platelet Involvement in the Pathophysiology of Infectious Endocarditis

It has now been clearly established that platelets are involved from the early stages of endocardial vegetation formation. Platelets possess a panoply of receptors on their surface that can interact with the bacteria involved in IE, with differences between species [10,13] (Table 1, Figure 1).

Platelets interact with *Staphylococcus aureus* via a direct contact involving the iron-responsive surface determinant B (IsdB) and glycoprotein IIb/IIIa on the *S. aureus* and platelet side, respectively, or between the staphylococcal accessory regulator protein and platelet GPIbα [14,15]. Platelets also interact with *S. aureus*, involving plasma proteins as bridges between the two protagonists, such as IgG between staphylococcal protein A and platelet FcγRII [16,17], fibrinogen or fibronectin between staphylococcal clumping factor A and B, or fibronectin-binding protein A and B and platelet GPIIbIIIa, among other platelet membrane receptors [18,19,20,21]. The bacterium can also induce platelet activation via bacterial molecules secreted into the microenvironment, such as α toxin and staphylococcal superantigen-like 5 (SSL-5) [22,23]. *S. aureus* is the pathogen most associated with embolic events [2]. In addition to its role in activating primary hemostasis, *S. aureus* has the capacity to directly initiate coagulation via its two coagulases, staphylocoagulase and vWbp, but also has the capacity to break down clots via its plasminogen activator staphylokinase, a potent activator of plasmin [24].

Platelet aggregation can also be induced by other Gram-positive bacterial species, such as streptococci and enterococci. With regards to streptococci, the mechanisms responsible for aggregation differ depending on the strain. Aggregation induced by *Streptococcus sanguinis* involves GPIIbIIIa and GPIb, through a VWF-independent mechanism [25], or through a serine-rich glycoprotein (SrpA) [26]. Other species belonging to the Streptococcus genus have also been shown to induce platelet aggregation. *Streptococcus gordonii* has a serine-rich repeat adhesin (SRR), like the *S. gordonii* surface proteins (GspB), which induce platelet activation by interacting with platelet GPIb [27]. As it is not the only mechanism, *S. gordonii* also induces platelet activation through the platelet adherence protein A (PadA), which specifically interacts with platelet GPIIbIIIa [28]. In addition, *Streptococcus agalactiae* binds to platelets via its SRR-1, using a fibrinogen bridge [29].

*Enterococcus faecalis* might also induce platelet aggregation via several mechanisms, which mainly are unknown today. The aggregation substance encoded by the plasmid pCF10 has been shown to induce platelet aggregation and represents a virulence factor in the promotion of IE [30]. Pilus Ebp represents another enterococcal virulence factor that has been involved in platelet aggregation during the formation of vegetation [31]. Enterococcal leucine-rich protein A (ElrA) appears to interact with VWF domains which could form a bridge between the bacteria and the host cells, including the platelets [32]. However, a difference between *E. faecalis* strains has been reported by Johansson et al., relating to their aggregating power, showing that some strains were able to aggregate platelets, while others were not [33]. Previously, Rasmussen et al. reported that *E. faecalis*-induced platelet aggregation differed between blood donors. Specific IgG binding to platelets would be involved in platelet activation by this bacterium [34]. The prophage dynamics of *E. faecalis* prophages contribute to pathogenicity by prophages pp1, pp4, and pp6 contributing to platelet adhesion [35]. Genomic analyses have elucidated significant inter- and intrastrain genomic microdiversity among *E. faecalis* isolated from IE heart valves. This microdiversity, which is expressed by mutations or complete deletions of virulence genes, plays a leading role in the pathophysiology and survival of the bacterium and has been found in one patient to correspond to the prophage pp4, which carries genes encoding the proteins PlbA and PlbB involved in platelet binding [36]. Colomer-Winiter et al. demonstrated for the first time the critical role of (p)ppGpp levels and its essential nature for *E. faecalis* colonization of the cardiac valve by characterizing the basal level of (p)ppGpp affecting the pathophysiological interaction of the bacterium [37].

IE can be caused by intracellular bacteria, such as *Coxiella burnetii* and *Tropheryma whipplei*, as well as facultative intracellular bacteria, such as the Bartonella genus. In these cases of IE, vegetation may be poorly developed, as in the case of *C. burnetii* endocarditis, or may be of appreciable size, as in the case of *T. whipplei* and Bartonella endocarditis [38,39,40,41]. Intuitively, we tend to think that no platelet–bacterium interaction can be initiated when the two cells are present in different environments. However, we have recently reported the case of a patient with *T. whipplei* IE, where large amounts of the bacterium were present in the extracellular compartment, in the presence of fibrin and platelet networks [42]. However, we do not know whether a platelet–*T. whipplei* interaction exists, and if so, via what mechanism.

**Table 1 ijms-24-07540-t001:** Involvement of bacterial membrane surface molecules in platelet activation.

Bacterial Surface Molecule	Plasma Protein	Platelet	Reference
*Staphylococcus aureus*			
IsdB	-	GPIIbIIIa	[14]
Sar P	-	GPIbα	[15]
Protein A	IgG	FcγRII	[16]
	VWF	GPIbα	[17]
Clf A and Clf B	Fg and Fn	GPIIbIIIa	[18,19,20,21]
FnBP			
*Viridans streptococci*			
SrpA	-	GPIbα	[26]
GspB	-	GPIbα	[27]
PadA	-	GPIIbIIIa	[28]
SRR-1	Fg	?	[29]
*Enterococcus faecalis*			
AS	-	?	[30]
Ebp	-	?	[31]

AS: aggregation substance; Clf: clumping factor; Ebp: endocarditis- and biofilm-associated pili; FcγR: crystallizable fragment gamma receptor; Fg: fibrinogen; Fn: fibronectin; FnBP: fibronectin-binding protein; GspB: Streptococcus gordonii surface platelet B; IgG: immunoglobulin G; IsdB: iron-responsive surface determinant B; PadA: platelet adherence protein A; Sar P: staphylococcal accessory regulator protein; SrpA: serine-rich glycoprotein A; SRR-1: serine-rich repeat glycoprotein 1; VWF: von Willebrand factor. ?: no data available.

Genetic polymorphisms have been found to play a significant role in the pathogenesis of IE. Toll-like receptors (TLRs) are a family of pattern recognition receptors (PRRs) that are constitutively expressed on immune cells and act as sentinels against exogenous and endogenous “danger” signals. These receptors interact with pathogen-associated molecular patters (PAMPs) and damage-associated molecular patterns (DAMPs) on the side of the microorganism or necrotic cells and trigger a series of molecular pathways that lead to the induction of innate immunity [43]. Activation of TLR signaling also induces maturation of dendritic cells (DCs), which is responsible for causing induction of the second line of host defense, known as adaptive immunity [44]. Furthermore, all TLRs have been found to be expressed in human cardiomyocytes and endothelial cells, and they have been implicated in the pathogenesis of several cardiovascular, inflammatory, and infectious diseases [45,46]. Specifically, the C/C genotype of the rs3775073 polymorphism in the TLR6 gene has been identified as a protective factor against IE [47], while the A allele of the rs5743708 polymorphism in the TLR2 gene has been suggested as a risk factor for the disease [48]. Among all TLRs identified to date, platelets express TLR1, TLR2, TLR4, and TLR6 with different expression levels. TLR2 and TLR4 are the most highly expressed on the platelet surface and induce intracellular activation by a MyD88-dependent mechanism [43]. TLR2 has been implicated in platelet aggregation induced by *Streptococcus agalactiae* [49]. Although TLRs are key receptors in immunological processes, little is known about their direct role in platelet–bacteria interactions involved in IE, namely with staphylococcal and enterococcal species.

Platelet parameters may play a role in prognostic assessment during the course of IE. One study reported a significant increase in mean platelet volume (MPV) in patients with IE and that this volume returned to normal following treatment [50]. In addition to making a similar observation, Cho et al. reported a significantly reduced platelet count and increased MPV/platelet count ratio in IE patients compared with the control group [51]. Furthermore, other studies have reported that, among patients with IE, MPV was even greater in patients with thrombotic events [52,53]. Finally, in their study published in 2020, Liu et al. revealed that the MPV/platelet count ratio was an independent predictor of increased risk for all-cause death [54].

It should be noted that MPV and platelet counts are not specific parameters for IE and that modifications to them are encountered in several pathologies [55,56]. It should also be noted that there are no limit values according to which the management of IE is decided. However, these results are promising, and further studies are needed to ensure the optimal use of these parameters in clinical practice in cases of IE.

## 3. Platelets Are Key Actor in the Formation of Vegetation

Although the pathophysiology of IE is still poorly understood, the role of platelets in the formation of vegetation has been clearly established. Histological staining of valves in suspected IE has been routinely performed for decades.

Early electron microscopy works detected the presence of platelets and fibrin in the vegetations [57]. In an earlier study, we demonstrated that the composition and cellular organization of endocardial vegetation differs according to the causative bacterial species [58]. Indeed, among the various vegetations analyzed, platelets and fibrin networks were abundant in cases of vegetations caused by *S. aureus* and streptococcus species, contrary to the vegetation due to *E. faecalis*, where the presence of erythrocytes was more marked.

Not only that, in their mouse-model-based study published in 2019, Liesenborghs et al. were able to establish a dichotomy of cellular organization during the initial phase of *S. aureus* adhesion, depending on whether the valve is injured or inflamed. Indeed, in patients with inflammation-induced endocarditis who develop a catheter infection, or in intravenous drug users, for example, VWF is released following activation of endothelial cells, leading to platelet adhesion to the valve surface. Thus, *S. aureus* adheres via platelets, which act as bridges in order to overcome the shear stress. Direct binding to VWF plays only a minor role in this process. In contrast, in injury-induced endocarditis, such as in rheumatic and congenital valve disease, where turbulent blood flow damages the endothelium, the subendothelium is exposed, resulting in local deposits of fibrin and VWF, to which *S. aureus* can directly adhere through adhesins such as VWF-binding protein (vWbp) and clumping factor (Clf). Direct binding to platelets plays only a minor role in this process. Since VWF is rapidly cleaved at the endothelial surface, its role could be to slow down the circulating bacteria, which then adhere to the subendothelium by additional adhesins, such as the collagen adhesin (Cna) of *S. aureus* which binds to collagen. [59].

In the same sense, in a case report on a teenager suffering from endocarditis on an inflamed valve following sepsis, we observed that *S. aureus* were localized in a necrotic region, probably formed of traces of platelets and inflammatory cells, as proposed by Liesenborghs et al. As the teenager’s vegetation was in a late stage, two other regions were characterized, namely an amorphous region resembling a biofilm and a cell-rich region [60].

Finally, in their review published in 2020, Liesenborghs et al. provided an explanation for the epidemiological changes in IE through this previously established dichotomy. Thus, in recent decades, rheumatic heart disease has become rare, with the emergence of new risk factors, such as valve prostheses, degenerative valve disease, and the use of the intravenous route. This has led to the emergence of more and more cases of structurally normal valve endocarditis at the expense of damaged valves. Inflammation of heart valves follows sepsis, intravenous drug use, or degenerative valve disease and is characterized by the widespread activation of endothelial cells. When endocarditis follows valve inflammation, *S. aureus* is the predominant pathogen [61].

## 4. Involvement of Platelets in Antibacterial Immunity

Although platelets are involved in bacterial adhesion to the heart valves, they may have a beneficial effect during IE through their bactericidal capacity. Thus, in a rabbit model, Sullam et al. reported that vegetations from thrombocytopenic rabbits weighed significantly less than those from controls. However, thrombocytopenic rabbits had both greater total amounts of bacteria within vegetations and higher densities of microorganisms within infected tissue [62]. Platelets contain molecules with antibacterial activity in their α granules, known as thrombin-induced platelet microbicidal proteins (tPMPs). Among other things, this concerns platelet factor 4 (PF4), as well as thrombocidins (TC1 and TC2), which initially play a role in hemostasis [63,64]. In addition, platelets secrete human β-defensin 1 (hBD-1), an extra granular molecule with bactericidal activity [65]. It has been suggested that these peptides increase the permeability of the bacterium membrane, leading to cell death. Indeed, after platelet activation, the bacterium membrane releases molecules into the environment that inhibit bacterial growth. This effect has been mainly demonstrated on *S. aureus*, but also on other bacterial species, such as *Escherichia coli* [66,67,68].

However, it has been shown that isolates from patients with IE were more often resistant to platelet microbicide proteins [69]. In addition, resistance to these proteins would make the bacteria much more virulent in experimental models [70].

Platelets also play a role in anti-infective immunity via chemotaxis of immune cells such as neutrophils and participate in the formation of neutrophil extracellular traps (NETs) via the interaction of several ligand–receptor pairs, especially P-selectin (CD62P) and P-selectin glycoprotein ligand 1 (PSGL-1) on the platelet and neutrophil sides, respectively [71,72]. Although NETs initially present a network to trap bacteria and thus prevent their growth and dissemination, it was revealed that NETs participate in the formation of immunothrombosis, with secondary effects on the host by inducing tissue damage through inducing the formation of vegetation, uncontrolled thrombosis, and subsequent inflammation [73,74]. Thus, platelets are important players in the pathogenesis of IEs. Although it is very complex to determine whether their different functions have a protective or vegetation promoting formation, Table 2 summarizes their main actions.

## 5. Therapeutics Involving Platelets

The obvious involvement of platelets in IE has piqued the interest of several teams regarding the possible benefit of antiplatelet drugs in the management of related thrombotic events. Veloso et al. have published two animal studies reporting that aspirin plus ticlopidine, the first antiplatelet agent targeting the platelet P_2_Y12 receptor, and abciximab, an anti-GPIIbIIIa antibody, had a statistically significant protective effect in rats against *S. aureus* and *Streptococcus gordonii* IE and that aspirin plus ticlopidine significantly reduced the weight of vegetation and protected 73% and 64% of rats (*p* < 0.005) from IE due to *E. faecalis* and *Streptococcus gallolyticus*, respectively [80,81].

Clinical studies have also looked at this issue, with aspirin attracting the most interest. However, discordant results have been reported. Anavekar et al. demonstrated that aspirin, when taken before the onset of IE, was associated with a significant reduction in embolic events [82]. However, the same author was unable to confirm this effect in a subsequent study [83]. Other studies have also reported no beneficial effect of aspirin on thrombotic events [84,85,86]. However, one study that only looked at *S. aureus* IE showed a statistically significant reduction with aspirin in univariate analysis, losing significance in multivariate analysis, although aspirin retained its role as a predictor of the reduced risk of acute valve surgery [87].

The studies mentioned above were retrospective studies, with the possibility of such biases as a difference in comorbidities between the two groups with and without aspirin. Only one prospective study, conducted by Chan et al., looked at the efficacy of aspirin in the management of IE. This study showed no significant difference and a tendency to bleed more under aspirin, when aspirin was started 30 days after the beginning of the infection [88]. These clinical studies evaluating the efficacy of aspirin in IE present several methodological differences, such as the daily dose of the drug and the time when treatment began, as well as the bacterial species involved.

With this in mind, in 2020 we conducted an in vitro study evaluating the efficacy of different antiplatelet molecules, namely, aspirin, ticagrelor, and the combination of aspirin–ticagrelor and tirofiban, on the aggregation induced by strains of *S. aureus* and *S. sanguinis*, using light transmission aggregometry (LTA). Through this study, we demonstrated that the efficacy of the medication differed depending not only on the species but also on the bacterial strain involved. Indeed, although aspirin showed a significant decrease in the platelet aggregation induced by most strains belonging to both species, ticagrelor showed the strongest inhibitory effect on aggregation induced by *S. aureus*, among the drugs taken orally. Meanwhile, for *S. sanguinis*, it was the combination of aspirin and ticagrelor which showed the strongest inhibitory effect [89] (Figure 2).

Apart from IE, one clinical study has recently shown that clopidogrel significantly reduced mortality from *S. aureus* bacteremia [90]. Other studies have also reported that ticagrelor reduced susceptibility to Gram-positive bacteria, mainly in pulmonary injuries and sepsis, in patients admitted for acute coronary syndrome [91,92,93,94]. One study attempted to explain the mechanics of such an effect. Indeed, it was shown that ticagrelor had a bactericidal effect on Gram-positive bacteria, including resistant strains, such as methicillin-resistant *S. aureus* (MRSA), glycopeptide-intermediate *S. aureus* (GISA), and vancomycin-resistant *E. faecalis* (VRE), in addition to methicillin-sensitive *S. aureus* (MSSA), *Staphylococcus epidermidis*, and *Streptococcus agalactiae* [95]. It should be noted that the minimum inhibitory concentration against these strains was obtained with supraphysiological concentrations. 

Taken together, these results show a possible benefit of antiplatelet agents in the management of IE. Further clinical studies are needed which take into consideration the bacterial strain involved as well as the time of initiation with respect to the onset of the infection.

Since vegetations are composed of, among other things, fibrin, and *S. aureus* has coagulase activity, previous studies have assessed the potential effect of anticoagulants such as vitamin K antagonists in the management of embolic events related to IE. These studies have reported conflicting results, in addition to some being linked to an increased risk of bleeding [96,97,98,99]. Nevertheless, dabigatran could be a potentially interesting choice. In addition to being a direct thrombin inhibitor, dabigatran also interferes with the coagulase activity of *S. aureus*. Indeed, *S. aureus* coagulases bypass the coagulation cascade to bind directly with prothrombin and form staphylothrombin which is directly active on fibrinogen. Thus, dabigatran would serve both as an anticoagulant and an inhibitor of bacterial virulence [100,101]. Preliminary experimental results are promising regarding the efficacy of dabigatran [102], however, further clinical studies are required to confirm its efficacy in humans.

In addition to drugs acting on hemostasis, statins have also shown a beneficial effect on thromboembolic events and mortality related to IE. Statins act by inhibiting 3-hydroxy-methylglutaryl co-enzyme A (HMG CoA) reductase and they are used to reduce cholesterol biosynthesis as well as in the primary and secondary prevention of cardiovascular diseases. In 2011, Anavekar et al., in their retrospective cohort, reported a significantly reduced rate of embolic events in statin users compared to non-statin users [83]. In another study published in 2014, it was reported that statins were associated with a reduced risk of in-hospital and subsequent mortality as a result of IE [103]. Previous studies have reported an antiplatelet effect of statins. Several hypotheses have been advanced to explain this effect, including an upregulation in nitric oxide synthesis by endothelial cells, resulting in an inhibitory effect on platelet activation. This raises the question of the magnitude of such an effect in IE, where the properties of the endothelium are often altered [104,105]. We attempted to assess the effect of three statin molecules, namely fluvastatin, atorvastatin, and rosuvastatin, on the antistaphylococcal effect of washed platelets in vitro. Interestingly, our results showed that all three molecules increased the antibacterial effect of platelets in a dose-dependent manner with an increase in CD41 and CD62P expression on the platelet surface [106]. Most recently, a new observational study reported that pre-admission and in-hospital statin use were associated with a lower risk of one-year mortality in patients with IE compared with those who did not use statins and those who had discontinued their use [107]. The explanation of such an effect is far from fully elucidated. Further, in vitro experimental studies and clinical trials are required to confirm the efficacy of statins in IE and to provide the complete mechanism of action for optimal use.

## 6. Major Research Techniques Used for Platelet–Bacteria Interaction Studies

The study of platelet–bacteria interactions is a crucial aspect of IE research to improve the understanding of vegetation formation. Various techniques have been employed to study these interactions. Light transmission aggregometry (LTA) can be performed on whole blood, platelet-rich plasma, or washed platelet samples using specialized equipment, such as a platelet aggregometer [108]. LTA, which is considered the gold standard for evaluating platelet function [109], is widely used in basic research to assess the impact of genetic disorders and therapeutic impact on platelet functions. Flow cytometry (FCM) is also used in routine practice to diagnose thrombopathies.

Besides their daily use for diagnosis, LTA and FCM have been used in this research field to characterize in vitro interactions between platelets and IE causative agents.

They have been used to assess the platelet activation and aggregation capacity of pathogens involved in IE, to evaluate the interest of antiplatelets, as well as for quantification of released platelet microparticles, which contribute to the spread of bacteria in the bloodstream [110]. We have previously studied the platelet response induced by the major bacteria causing IE and demonstrated the heterogeneity of the platelet response induced by these strains [67,68,89].

Proteomics and immunological techniques have also been used to investigate such interactions. The use of fluorescently labeled antibodies has improved the accuracy and reproducibility of platelet proteomics studies. The measurements of platelet activation soluble markers (platelet secretome) and membrane markers (platelet sheddome) through proteomic techniques have provided valuable information on the role of platelets in immune defense against bacteria [111]. Recently, in our reference center, a specific Western blot test has been developed to diagnose the main causative agents in the case of blood culture negative endocarditis (BCNE) [112]. Transcriptomics and genomics techniques can also be used to study platelet–bacteria interactions. RNA-seq microarrays and coding transcriptome analysis provide global molecular insights into gene expression, while pathway analysis and expression identification help to better understand the regulatory mechanisms involved in platelet production. RT-qPCR and gene expression analysis can be used for the diagnosis of heritable platelet disorders, mutation detection, and identification of genes associated with platelet function [113]. We estimate that these methods could be an interesting approach to characterize platelet/bacterial genes implied in IE.

The study of platelet biology has been significantly advanced using imaging tools, including fluorescence and phase contrast microscopy. Transmission electron microscopy (TEM) and scanning electron microscopy (SEM) are two advanced techniques used in the study of platelets and their interactions with pathogens, such as bacteria [114]. TEM offers a detailed view of platelet granules, platelet defects, and organelle dynamics at high magnification [115], while SEM provides a high-resolution surface view of platelets and their behavior when they are in contact with different pathogens [116]. Moreover, we developed a new approach for the analysis of the heart valve ultrastructure of IE patients by combining SEM and energy-dispersive X-ray spectroscopy which allowed us to characterize the chemical profile of the cardiac valves infected with *Viridans streptococci* as well as the arrangement of the different cells on the infected valve [58]. 

In complement, confocal imaging is a powerful tool for studying the behavior and morphology of platelets and bacteria in real time and provides valuable information about platelet spreading, protein localization, and cytoskeletal changes [111]. Prudent et al. used, for the first time, fluorescence in situ hybridization (FISH) with peptide nucleic acid probes (PNA) for detection of *Coxiella burnetii* in heart valves from patients with IE [117]. We suggest that a combination of this technique and immunofluorescence could be considered for a higher sensitivity and specificity as well as to better visualize the distribution of cells and proteins involved in the heart valve in the IE context.

Animal models have played a crucial role in advancing our understanding of the pathogenesis and host defense mechanisms of IE. Various animal models, including mice, rats, rabbits, and pigs, have been developed to study IE and have highlighted the significance of bacterial virulence factors such as adhesion [59], biofilm formation [73,118], and toxin production [119] in the development of the disease. While these models have been invaluable in advancing our knowledge, they have limitations in elucidating the direct role of platelets in the pathogenesis of IE. However, early studies have shown that platelets play a critical role in the development of streptococcal endocarditis in rabbit and rat models [62,120]. Additionally, studies of rat models have demonstrated that bacteria can stimulate the formation of vegetation by inducing intravascular NETs via activated platelets [71,73]. Animal experimentation has also contributed to the determination of the effect of antiplatelet agents [81,82,121]. Nevertheless, the clinical use of these agents is not yet supported [122].

Despite the valuable information provided by animal models, they still fail to reproduce the complexity of the disease due to inherent limitations. Although pigs and rabbits have cardiovascular and immune systems very similar to those of humans, they remain imperfect surrogates.

In addition to the different techniques available, advances need to be made in the development of various multidisciplinary methods for examining platelets. The variety of multidisciplinary techniques developed by combining available techniques allows the researcher to select the most appropriate measurement method depending on the specific research objective. With the growing body of evidence showing the crucial role of platelets in the prognosis of not only IE, but also diseases involving platelets previously ignored, these technologies open an exciting new era for platelet studies, and some of them could potentially be used in medical practice.

## 7. Conclusions

In conclusion, new concepts in the pathogenesis of IE have emerged in recent years, notably the origin of inflammation or damage triggering different mechanisms. Moreover, the discovery of new roles for platelets, including antibacterial and immunological roles, in addition to the classic hemostatic role, as well as the different behaviors of bacterial strains towards platelets, should open new avenues towards understanding the pathophysiological mechanisms of this disease. The data currently available indicate the presence of different IEs, depending on both the host and the infectious agent in question. Finally, several drug classes are candidates for the prevention of thromboembolic events related to IE, including antiplatelet agents, dabigatran, and statins. Further clinical trials are needed to discern their efficacy and potentially make them available to patients. The new diagnosis tools might allow us to confirm our hypothesis and to evaluate the potential effect of antithrombotic drugs.

## Figures and Tables

**Figure 1 ijms-24-07540-f001:**
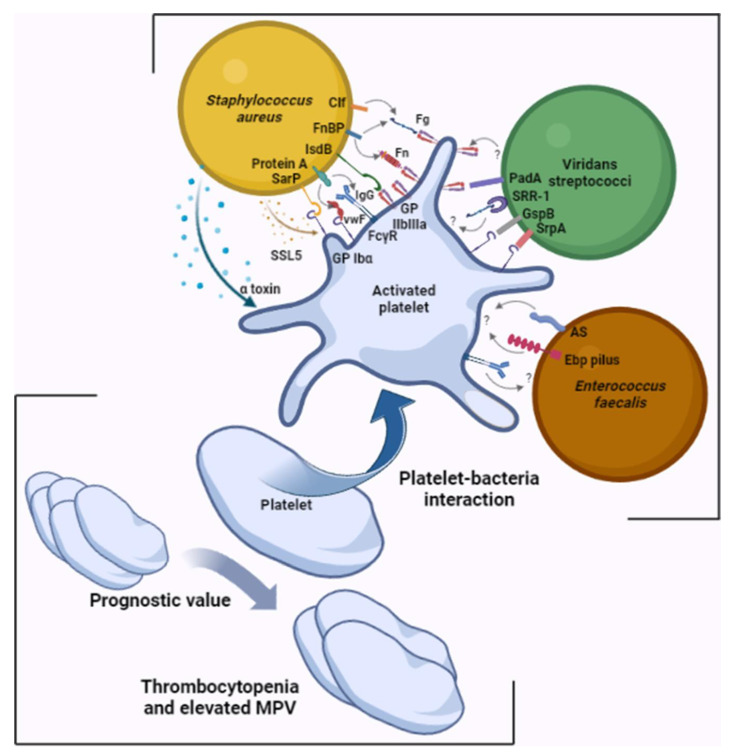
Platelets, a cornerstone of infective endocarditis. Different interactions between platelets and bacterial species involved in IE involving numerous ligand–receptor pairs and predictive value of changes in platelet parameters. Illustrations were created using https://www.biorender.com, accessed on 17 March 2023. AS: aggregation substance; Clf: clumping factor; Ebp: endocarditis- and biofilm-associated pili; FcγR: crystallizable fragment gamma receptor; Fg: fibrinogen; Fn: fibronectin; FnBP: fibronectin-binding protein; GspB: Streptococcus gordonii surface platelet B; IgG: immunoglobulin G; IsdB: iron-responsive surface determinant B; PadA: platelet adherence protein A; Sar P: staphylococcal accessory regulator protein; SrpA: serine-rich glycoprotein A; SRR-1: serine-rich repeat glycoprotein 1; SSL5: staphylococcal superantigen-like 5; VWF: von Willebrand factor.

**Figure 2 ijms-24-07540-f002:**
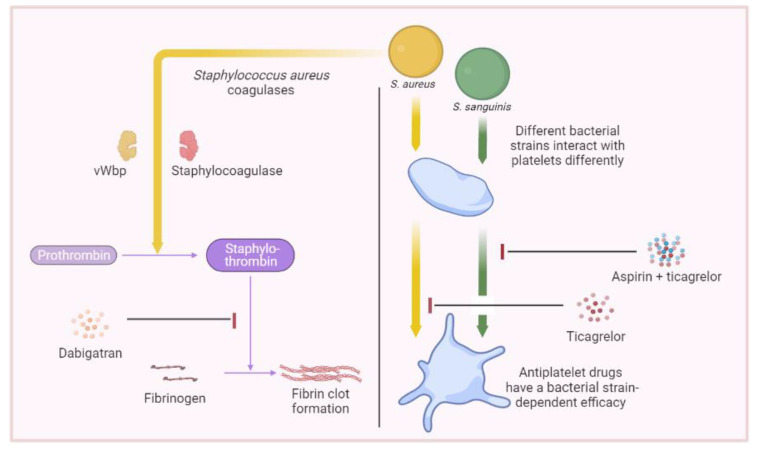
The effect of antithrombotic agents on hemostasis induced by bacteria. **Right**, the effect of oral antiplatelet agents on platelet aggregation induced by different bacterial strains. **Left**, the effect of dabigatran on coagulation induced by *Staphylococcus aureus* coagulases. Illustrations were created using https://www.biorender.com, accessed on 17 March 2023. vWbp: von Willebrand factor-binding protein.

**Table 2 ijms-24-07540-t002:** Main functions of platelets during IE.

Platelet aggregation, platelet–leukocyte aggregation, activation of coagulation	[75]
Secretion of bactericidal substances contained in their alpha granules and mechanical removal of bacteria	[63,64,65,76]
Activation of NETosis	[71,72,73,74,77]
Synthesis by platelets of immune mediators for the activation of T and B lymphocytes, macrophages, and dendritic cells	[78]
Activation of complement system	[75]
Activation of endothelium and extravasation	[79]

## Data Availability

Not applicable.

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
