# Peer review of "The Role of Platelets in Infective Endocarditis"

_ijms, 2023, doi:10.3390/ijms24087540_

Round 1
Reviewer 1 Report
Braï and colleagues provide a review of current literature describing the intricate interplay of platelets in infective endocarditis. Platelet involvement in the pathology, formation of vegetation, and as agents that disperse anti-bacterial properties are discussed. Strategies that target platelets, and research techniques to study infective endocarditis are also explored. The authors have compiled a mix of articles ranging from clinical, through animal models and with isolated blood components. The document is informative, well-written, and ultimately highlights work recently published from the same group.
Main concerns
This review would be strengthened with the addition of a more comprehensive introduction. I would suggest including a discussion of the clinical diagnostic tests performed for patients, and then transition into the common animal or in vitro model systems used.
The section titled “Platelet involvement in the pathophysiology…” covers quite a bit of literature. It would strengthen the review to consider generating a table that summarizes a portion of that information. I will note that Figure 1 does attempt this, but inclusion of a table will both highlight the information further and readily bring readers to the source material.
One major mechanism of pathogen recognition and interaction between bacteria and platelets is through toll-like receptors (TLRs). In an effort to make the manuscript more comprehensive, I would suggest highlighting any relevant studies on TLRs in infective endocarditis.
Please clarify the sentence on lines 195-197. The overall message of the sentence itself is unclear, and the citation appears to be referencing a different study.
Please clarify ‘sheddome’ in line 330. Is this referring to shedding of cell receptors with age, or release of extracellular vesicles?
Minor Concerns
Please note the redundancy in line 66 and 152.
In Figure 1. Thrombocytopenia should replace thrombocytomenia
Author Response
Pr Laurence Camoin-Jau
Service d’Hématologie Biologique
Assistance Publique Hôpitaux de Marseille
Centre Hospitalier Universitaire Timone
164 Rue Saint Pierre
13005 Marseille
France
Courriel: Laurence.camoin@ap-hm.fr
Marseille, April 2nd 2023
Object : Manuscript ID: ijms-2288499
Dear reviewer
We have responded to all of your comments. Please check the new version of the manuscript. Corrections are in bold in the text. We are of course at your disposal to answer any criticism and remarks.
Please be assured that we are grateful for your trust.
Sincerely yours
Pr L Camoin-Jau
Reviewer 1
Open Review
(x) I would not like to sign my review report
( ) I would like to sign my review report
Quality of English Language
( ) English very difficult to understand/incomprehensible
( ) Extensive editing of English language and style required
( ) Moderate English changes required
(x) English language and style are fine/minor spell check required
( ) I am not qualified to assess the quality of English in this paper
|
Is the work a significant contribution to the field? |
|
|
Is the work well organized and comprehensively described? |
|
|
Is the work scientifically sound and not misleading? |
|
|
Are there appropriate and adequate references to related and previous work? |
|
|
Is the English used correct and readable? |
Comments and Suggestions for Authors
Braï and colleagues provide a review of current literature describing the intricate interplay of platelets in infective endocarditis. Platelet involvement in the pathology, formation of vegetation, and as agents that disperse anti-bacterial properties are discussed. Strategies that target platelets, and research techniques to study infective endocarditis are also explored. The authors have compiled a mix of articles ranging from clinical, through animal models and with isolated blood components. The document is informative, well-written, and ultimately highlights work recently published from the same group.
Main concerns
- This review would be strengthened with the addition of a more comprehensive introduction. I would suggest including a discussion of the clinical diagnostic tests performed for patients, and then transition into the common animal or in vitro model systems used.
Response
As expected, a more comprehensive introduction was added line 42.
The occurrence of bacteremia on a pathological valve or foreign material results in complex interactions between the microorganism, the valve endothelium, and host immune responses. Normal valve endothelium is resistant to thrombosis and infection. However, its alteration due to native valve disease exposes the extracellular matrix components of the subendothelium which is thrombogenic and trigger platelet adhesion and activation, followed by rapid colonization of circulating microorganisms [1]. Differences in the affinity of microorganisms for the valve depend on adhesion molecules (MSCRAMM: microbial surface component reacting with adhesive matrix molecules). Fibrinogen- or fibronectin-binding proteins are present on the surface of microorganisms. Some microorganisms such as S. aureus can also bind directly to endothelial cells. The properties of these numerous adhesion molecules explain the predominance of gram-positive cocci as the responsible microorganisms [7]. The particular adhesion capabilities of S. aureus may explain its predominance in endocarditis, occurring in the absence of pre-existing valvulopathy [2]. Adhesion of microorganisms is also promoted by integrin expression by endothelial cells in response to inflammation. After the adhesion phase, microorganisms can be internalized by endothelial cells where they can proliferate and diffuse, or persist by evading antibiotics and the immune system. The mechanisms of colonization of the valve endothelium differ and are poorly understood for microorganisms with exclusive intracellular development, such as Coxiella burnetii and Bartonella.
Some microorganisms, particularly S. aureus, accompany their proliferation with biofilm formation leading to platelet aggregation embedded in a polysaccharide and protein network. Biofilm formation promotes virulence of the bacterium, particularly by protecting them from immune defenses and antimicrobial treatments [8]. Colonization of the fibrino-platelet thrombus by circulating microorganisms triggers an immune response, activating monocytes that secrete tissue factors and cytokines. The invasion of the valve tissue by microorganisms and then inflammatory cells is the cause of the main lesions and complications of IE. The vegetations correspond to a septic thrombus, whose growth is promoted by the activation of haemostasis by cytokines. Microorganisms that have colonized the vegetations are difficult to access by antibiotics and immune defenses [9].”
Moreover, as expected, discussion of the clinical diagnostic tests performed for patients, and then transition into the common animal or in vitro model systems used was added ( line 420).
“Animal models have played a crucial role in advancing our understanding of the pathogenesis and host defense mechanisms of IE. Various animal models, including mice, rats, rabbits, and pigs, have been developed to study IE and have highlighted the significance of bacterial virulence factors such as adhesion [59], biofilm formation [73, 118], and toxin production [119] in the development of the disease. While these models have been invaluable in advancing our knowledge, they have limitations in elucidating the direct role of platelets in the pathogenesis of IE. However, early studies have shown that platelets play a critical role in the development of streptococcal endocarditis in rabbit and rat models [62, 120]. Also, studies of rat models have demonstrated that bacteria can stimulate the formation of vegetation by inducing intravascular NETs via activated platelets [71, 73]. Animal experimentation has also contributed to the determination of the effect of antiplatelet agents [81, 82, 121]. Nevertheless, the clinical use of these agents is not yet supported [121].
Despite the valuable information provided by animal models, they still fail to reproduce the complexity of the disease due to inherent limitations. Although pigs and rabbits have cardiovascular and immune systems very similar to those of humans, they remain imperfect surrogates.
In addition to the different techniques available, advances need to be made in the development of various multidisciplinary methods for examining platelets. The variety of multidisciplinary techniques developed by combining available techniques allows the researcher to select the most appropriate measurement method depending on the specific research objective. With the growing body of evidence showing the crucial role of platelets in the prognosis of not only IE, but also diseases involving platelets previously ignored, these technologies open an exciting new era for platelet studies, and some of them could potentially be used in medical practice.”
- The section titled “Platelet involvement in the pathophysiology…” covers quite a bit of literature. It would strengthen the review to consider generating a table that summarizes a portion of that information. I will note that Figure 1 does attempt this, but inclusion of a table will both highlight the information further and readily bring readers to the source material.
Response:
Dear Editor, thank you for your suggestion. We have thus introduced a table summarizing the involvement of bacterial membrane surface molecules in platelet activation. Please check table 1 in the new submitted version.
Table 1 : involvement of bacterial membrane surface molecules in platelet activation
AS: aggregation substance; Clf: clumping factor; Ebp: endocarditis and biofilm associated pili; FcγR: Crystallisable fragment gamma receptor; Fg: fibrinogen; Fn: fibronectin; FnBP: fibronectin binding protein; GspB: Streptococcus gordonii surface platelet B; IgG: Immunoglobulin G; IsdB: iron-responsive surface determinant B; PadA: platelet adherence protein A; Sar : Staphylococcal accessory regulator protein; SrpA: serin-rich glycoprotein A; SRR-1: serine-rich repeat glycoprotein 1; VWF: Von Willebrand Factor.
- One major mechanism of pathogen recognition and interaction between bacteria and platelets is through toll-like receptors (TLRs). In an effort to make the manuscript more comprehensive, I would suggest highlighting any relevant studies on TLRs in infective endocarditis.
Dear reviewer,
Although TLRs are platelet receptors highly involved in the platelet response, few data are available in IE models.
However, at your request, we have added a paragraph on these receptors, line 172.
“Genetic polymorphisms have been found to play a significant role in the pathogenesis of IE. Toll-like receptors (TLRs) are a family of pattern recognition receptors (PRRs) that are constitutively expressed on immune cells and act as sentinels against exogenous and endogenous "danger" signals. These receptors interact with pathogen-associated molecular patters (PAMPs) and damage-associated molecular patterns (DAMPs) on the side of the microorganism or necrotic cells and trigger a series of molecular pathways that lead to the induction of innate immunity [43]. Activation of TLR signaling also induces maturation of dendritic cells (DCs), which is responsible for alerting induction of the second line of host defense, known as adaptive immunity [44]. Furthermore, all TLRs have been found to be expressed in human cardiomyocytes and endothelial cells, and they have been implicated in the pathogenesis of several cardiovascular, inflammatory, and infectious diseases [45, 46]. Specifically, the C/C genotype of the rs3775073 polymorphism in the TLR6 gene has been identified as a protective factor against IE [47], while the A allele of the rs5743708 polymorphism in the TLR2 gene has been suggested as a risk factor for the disease [48]. Among all TLRs identified so far, platelets express TLR1, TLR2, TLR4, and TLR6 with different expression levels. TLR2 and TLR4 are the most highly expressed on the platelet surface and induce intracellular activation by a MyD88-dependent mechanism [43]. TLR2 has been implicated in platelet aggregation induced by Streptococcus agalactiae [49]. Although TLRs are key receptors in immunological processes, little is known about their direct role in platelet-bacteria interactions involved in IE, namely with staphylococcal and enterococcal species.”
- Please clarify the sentence on lines 195-197. The overall message of the sentence itself is unclear, and the citation appears to be referencing a different study.
Response:
Dear reviewer, thank you for your comment. It helped us to clarify the idea. We have reworded the sentence. Please see the new submitted version : lines 250-254.
“In a rabbit model, Sullam et al., reported that vegetations from thrombocytopenic rabbits weighed significantly less than those from controls. However, thrombocytopenic rabbits had both greater total amounts of bacteria within vegetations and higher densities of microorganisms within infected tissue [62].”
- Please clarify ‘sheddome’ in line 330. Is this referring to shedding of cell receptors with age, or release of extracellular vesicles?
Response
Dear reviewer,
The sentence was clarified (line 388).
“The measurement of platelet activation soluble markers, (platelets secretome), and membrane markers (platelets sheddome) through proteomic techniques have provided valuable information for role of platelets in immune defense against bacteria (Arbesu et al., 2016) ».
Minor Concerns
Please note the redundancy in line 66 and 152.
In Figure 1. Thrombocytopenia should replace thrombocytomenia
Response
Dear reviewer, we apologize for this oversight. We have corrected figure 1. Please check the new submitted version.
Submission Date
02 March 2023
Date of this review
13 Mar 2023 21:50:53

Reviewer 2 Report
This is a timely and interesting review, given the interest in the role and activation of platelets in inflammation and host defence mechanisms. The authors need to address the following points:
1. The authors need to provide conceptual understanding or hypothesis of the role of platelets in endocarditis, and for this to be considered over the continuum of the condition if they believe the functions of platelets change. For example compare the protective versus pathological consequences of platelet activation and function. This might be nicely summarised in a table and would give clearer insight.
2. The section on the use of anti-platelet drugs is necessary, but currently limited and needs expanding. There is little mention of what effect these drugs have on the immune response to overcome infection. Is inhibition of platelet activation a good or bad thing? Again, this needs answering with respect to the different functions of platelets, for example aggregation is not the same function as bacteriacidal activity, vascular integrity etc. Is there scope for considering modulating platelet activation via new targets that are not associated with haemostasis, but rather host defence?
3. The section on major research techniques is interesting, and could be expanded to include the development of animal models to address platelet activation and function over the time course of endocarditis.
Author Response
Pr Laurence Camoin-Jau
Service d’Hématologie Biologique
Assistance Publique Hôpitaux de Marseille
Centre Hospitalier Universitaire Timone
164 Rue Saint Pierre
13005 Marseille
France
Courriel: Laurence.camoin@ap-hm.fr
Marseille, April 2nd 2023
Object : Manuscript ID: ijms-2288499
Dear reviewer
We have responded to all of your comments. Please check the new version of the manuscript. Corrections are in bold in the text. We are of course at your disposal to answer any criticism and remarks.
Please be assured that we are grateful for your trust.
Sincerely yours
Pr L Camoin-Jau
Reviewer 2
Open Review
(x) I would not like to sign my review report
( ) I would like to sign my review report
Quality of English Language
( ) English very difficult to understand/incomprehensible
( ) Extensive editing of English language and style required
( ) Moderate English changes required
(x) English language and style are fine/minor spell check required
( ) I am not qualified to assess the quality of English in this paper
|
Is the work a significant contribution to the field? |
|
|
Is the work well organized and comprehensively described? |
|
|
Is the work scientifically sound and not misleading? |
|
|
Are there appropriate and adequate references to related and previous work? |
|
|
Is the English used correct and readable? |
Comments and Suggestions for Authors
This is a timely and interesting review, given the interest in the role and activation of platelets in inflammation and host defence mechanisms. The authors need to address the following points:
- The authors need to provide conceptual understanding or hypothesis of the role of platelets in endocarditis, and for this to be considered over the continuum of the condition if they believe the functions of platelets change. For example compare the protective versus pathological consequences of platelet activation and function. This might be nicely summarised in a table and would give clearer insight.
Response
Dear reviewer,
In our review, we described the interactions between platelets and bacterial species involved in infective endocarditis and also gave the results of studies that tested drugs acting on platelet action. You raise a very complex issue. Platelets have very multiple and complex roles in the course of EI. It is extremely difficult to compare the protective versus pathological consequences of platelet activation and function. Some functions have a protective role but also participate in the pathogenesis of the disease. Thus, we propose a table that presents the main functions of platelets in this pathology (Table 3).
- The section on the use of anti-platelet drugs is necessary, but currently limited and needs expanding. There is little mention of what effect these drugs have on the immune response to overcome infection. Is inhibition of platelet activation a good or bad thing? Again, this needs answering with respect to the different functions of platelets, for example aggregation is not the same function as bacteriacidal activity, vascular integrity etc. Is there scope for considering modulating platelet activation via new targets that are not associated with haemostasis, but rather host defence?
Response:
Dear reviewer, we thank you for your questioning which in our opinion requires further studies.
In our current review, we have discussed the effect of antiplatelet agents through the studies carried out so far, both in vitro, animal and clinical studies. Indeed, the effect of antiplatelet agents on the evolution of infective endocarditis revolves mainly around the inhibition of platelet aggregation. Furthermore, we have already evaluated the effect of these drugs on the antibacterial capacity of platelets and we have addressed this study in our current manuscript. The current treatment of infective endocarditis is based on the use of antibiotics, which could make up for the decrease in the antibacterial capacity of platelets by antiplatelet agents.
To our knowledge, there are still no clear answers to the question of the importance of antiplatelet agents in modulating platelet immunological responses in IE or the presence of molecules that only target aggregation without affecting host defense. Future studies in this area are highly recommended.
- The section on major research techniques is interesting, and could be expanded to include the development of animal models to address platelet activation and function over the time course of endocarditis.
Response
Dear reviewer,
As expected, we have developed a paragraph about the interest of animal model to address platelet activation and function over the time course of IE (line 420).
“Animal models have played a crucial role in advancing our understanding of the pathogenesis and host defense mechanisms of IE. Various animal models, including mice, rats, rabbits, and pigs, have been developed to study IE and have highlighted the significance of bacterial virulence factors such as adhesion [59], biofilm formation [73, 118], and toxin production [119] in the development of the disease. While these models have been invaluable in advancing our knowledge, they have limitations in elucidating the direct role of platelets in the pathogenesis of IE. However, early studies have shown that platelets play a critical role in the development of streptococcal endocarditis in rabbit and rat models [62, 120]. Also, studies of rat models have demonstrated that bacteria can stimulate the formation of vegetation by inducing intravascular NETs via activated platelets [71, 73]. Animal experimentation has also contributed to the determination of the effect of antiplatelet agents [81, 82, 121]. Nevertheless, the clinical use of these agents is not yet supported [121].
Despite the valuable information provided by animal models, they still fail to reproduce the complexity of the disease due to inherent limitations. Although pigs and rabbits have cardiovascular and immune systems very similar to those of humans, they remain imperfect surrogates.
In addition to the different techniques available, advances need to be made in the development of various multidisciplinary methods for examining platelets. The variety of multidisciplinary techniques developed by combining available techniques allows the researcher to select the most appropriate measurement method depending on the specific research objective. With the growing body of evidence showing the crucial role of platelets in the prognosis of not only IE, but also diseases involving platelets previously ignored, these technologies open an exciting new era for platelet studies, and some of them could potentially be used in medical practice”.
